A variant of Runx2 that differs from the bone isoform in its splicing is expressed in spermatogenic cells

Kanto Satoru 1 2 s.kanto@yamatoku-hp.jp
Grynberg Marcin 3 4
Kaneko Yoshiyuki 5
Fujita Jun 5
Satake Masanobu 1
1 Department of Molecular Immunology, Institute of Development, Aging and Cancer, Tohoku University , Sendai, Miyagi , Japan
2 Department of Urology, Graduate School of Medicine, Tohoku University , Sendai, Miyagi , Japan
3 Institute of Biochemistry and Biophysics, Polish Academy of Sciences , Warsaw , Poland
4 Program in Bioinformatics and Systems Biology, Stanford Burnham Medical Research Institute , La Jolla, CA , United States of America
5 Department of Clinical Molecular Biology, Faculty of Medicine, Kyoto University , Kyoto , Japan
Frauscher Ferdinand
Electronic publication date: 2016 Apr 4
Publication date: 2016
Volume: 4
Electronic Location ID: e1862
Received 2016 Jan 7; Accepted 2016 Mar 9
Copyright: ©2016 Kanto et al.
Copyright year: 2016
Copyright holder: Kanto et al.
License: This is an open access article distributed under the terms of the Creative Commons Attribution License, which permits unrestricted use, distribution, reproduction and adaptation in any medium and for any purpose provided that it is properly attributed. For attribution, the original author(s), title, publication source (PeerJ) and either DOI or URL of the article must be cited.
License URL: https://creativecommons.org/licenses/by/4.0/

Keywords: Alternative splicing, Cell differentiation, Spermatogenesis, Runx transcription factor

Funding: Ministry of Education, Science, Sports, Culture and Technology, Japan This work was supported by Grants-in-Aid from the Ministry of Education, Science, Sports, Culture and Technology, Japan.The funders had no role in study design, data collection and analysis, decision to publish, or preparation of the manuscript.

==============================
Background. Members of the Runx gene family encode transcription factors that bind to DNA in a sequence-specific manner. Among the three Runx proteins, Runx2 comprises 607 amino acid (aa) residues, is expressed in bone, and plays crucial roles in osteoblast differentiation and bone development. We examined whether the Runx2 gene is also expressed in testes.

Methods. Murine testes from 1-, 2-, 3-, 4-, and 10-week-old male mice of the C57BL/6J strain and W∕Wv strain were used throughout the study. Northern Blot Analyses were performed using extracts form the murine testes. Sequencing of cDNA clones and 5′-rapid amplification of cDNA ends were performed to determine the full length of the transcripts, which revealed that the testicular Runx2 comprises 106 aa residues coding novel protein. Generating an antiserum using the amino-terminal 15 aa of Runx2 (Met1 to Gly15) as an antigen, immunoblot analyses were performed to detect the predicted polypeptide of 106 aa residues with the initiating Met1. With the affinity-purified anti-Runx2 antibody, immunohistochemical analyses were performed to elucidate the localization of the protein. Furthermore, bioinformatic analyses were performed to predict the function of the protein.

Results. A Runx2 transcript was detected in testes and was specifically expressed in germ cells. Determination of the transcript structure indicated that the testicular Runx2 is a splice isoform. The predicted testicular Runx2 polypeptide is composed of only 106 aa residues, lacks a Runt domain, and appears to be a basic protein with a predominantly alpha-helical conformation. Immunoblot analyses with an anti-Runx2 antibody revealed that Met1 in the deduced open reading frame of Runx2 is used as the initiation codon to express an 11 kDa protein. Furthermore, immunohistochemical analyses revealed that the Runx2 polypeptide was located in the nuclei, and was detected in spermatocytes at the stages of late pachytene, diplotene and second meiotic cells as well as in round spermatids. Bioinformatic analyses suggested that the testicular Runx2 is a histone-like protein.

Discussion. A variant of Runx2 that differs from the bone isoform in its splicing is expressed in pachytene spermatocytes and round spermatids in testes, and encodes a histone-like, nuclear protein of 106 aa residues. Considering its nuclear localization and differentiation stage-dependent expression, Runx2 may function as a chromatin-remodeling factor during spermatogenesis. We thus conclude that a single Runx2 gene can encode two different types of nuclear proteins, a previously defined transcription factor in bone and cartilage and a short testicular variant that lacks a Runt domain.

Introduction

Runx transcription factors are characterized by the presence of a Runt domain (Kagoshima et al., 1993). This domain contains 130 amino acids (aa) and is responsible for sequence-specific DNA-binding activity and for dimerization with the protein PEBP2β∕CBFβ. There are three known Runx genes in mammals, Runx1, Runx2, and Runx3, each of which plays important roles in cell proliferation and differentiation as well as in the occurrence of specific human diseases (Wong et al., 2011; Chuang, Ito & Ito, 2013).

Runx2, the topic of this study, is expressed in bone, thymus, testis, and brain (Satake et al., 1995; Komori et al., 1997; Otto et al., 1997; Jeong et al., 2008). In bone and thymus, the Runx2 transcript contains a Runt domain sequence and the translated product functions as a transcription factor. In bone, gene-targeting studies have demonstrated that Runx2 is essential for the differentiation of immature osteoblasts into mature osteocytes. In mice lacking the Runt domain of Runx2, ossification of the skeletal system is severely impaired and the animals die soon after birth due to a respiratory defect (Komori et al., 1997). Of clinical importance, heterozygous loss of Runx2 causes cleidocranial dysplasia in humans, which is characterized by hypoplasia/aplasia of the clavicles and fontanelles (Otto et al., 1997; Mundlos et al., 1997).

In the thymus, Runx2 appears to function as an oncogene because the insertion of a retroviral genome near to the Runx2 locus in mice results in its overexpression and subsequently the occurrence of T-cell leukemia (Stewart et al., 1997). In addition, overexpression of a Runx2 transgene in the T-cell lineage perturbs the differentiation of thymocytes, mainly at the β selection stage, and produces a population that predominantly consists of immature CD8+ thymocytes (Vaillant et al., 2002).

Runx2 is also expressed in the testis. This was originally reported by Satake et al., (1995) and subsequently by Ogawa et al., (2000). According to Ogawa et al. (2000), the testicular Runx2 transcript displays several unique features. First, it is remarkably shorter (∼1.8 kb) than the transcripts found in bone (6.3 and 7.4 kb), mainly due to the premature termination of the testicular transcript within exon 8. Second, as a result of alternative splicing and fusion between exons 1 and 3, a new stop codon is generated in exon 3. The deduced open reading frame (ORF) encodes a polypeptide of only 106 aa residues. In addition, there are two methionine codons within exon 1 of this ORF, Met1 and Met69. Ogawa et al. (2000) predicted that Met69 is the translation initiation codon because the nucleotide sequence adjacent to Met69 is in better agreement with Kozak’s rule than the sequence adjacent to Met1 (Kozak, 2002). However, if Met69 was the start codon, then the encoded polypeptide would only be 38 aa residues long. Furthermore, because the alternative splicing removes exon 2, which encodes the amino-terminal portion of the Runt domain, the testicular Runx2 transcript cannot encode a Runt domain-containing transcription factor.

In this study, we investigated the possibility that Met1 rather than Met69 is used as the initiation codon for the translation of the testicular Runx2 transcript because the environment for translation in testicular cells is distinct from that in somatic cells. Furthermore, we examined the expression pattern of the putative 106-aa polypeptide in relation to the differentiation stages of testicular germ cells. We propose that the single Runx2 gene can encode two distinct types of protein: a small protein expressed in the testis that lacks a Runt domain, and a previously defined Runt-containing transcription factor that is expressed in bone and thymus.

Materials and Methods

Mice were maintained in the Animal Facility of the Institute of Development, Aging, and Cancer, Tohoku University, an environmentally controlled and specific pathogen–free facility. Animal protocols were reviewed and approved by the Animal Studies Committee of the Tohoku University (relevant approval number: 2013-IDAC-Animal-013).

Northern blot analysis

Testes were isolated from 1-, 2-, 3-, 4-, and 10-week-old male mice of the C57BL/6J strain and from 10-week-old male mice of the W∕Wv strain. Spermatocyte and spermatid fractions were prepared from the cell suspension of C57BL/6J testes (Mays-Hoopes et al., 1995). Total cytoplasmic RNA was prepared from testes using Isogen (Nippon Gene, Toyama, Japan). Poly(A)+ RNAs were selected using Oligo(dT)-Latex (Takara, Otsu, Japan) and 2 µg of sample was electrophoresed through a 1% (w/v) agarose gel containing 2.2 M formaldehyde. RNA was transferred from the gel to a membrane, and the membrane was hybridized with a 32P-labeled probe as described previously (Chiba et al., 1997). The probes were prepared either from a HindIII-NotI fragment of murine Runx2 cDNA (corresponding to nt 282 through to nt 473 in NM_001146038.2) or murine Runx1 cDNA sequence, murine PEBP2β∕CBFβ cDNA sequence and β-actin (Satake et al., 1995).

cDNA cloning and sequencing

A cDNA library prepared from murine testicular poly(A)+ RNA was provided by Y Nishina (Osaka University, Osaka, Japan). A 32P-labeled HindIII-NotI fragment of murine Runx2 cDNA was used as the probe. The library was screened under a stringent condition according to the standard method. The cDNA inserts from each of the five isolated clones were excised from the pAP3neo vector and subcloned into the pBluescript II vector. The entire length of the insert was sequenced using the dideoxy-dye terminator method.

5′-Rapid Amplification of cDNA Ends (5′-RACE)

To determine the full length of the transcripts, 5′-RACE was performed following the manual supplied by the manufacturer (Life Technologies). A gene-specific primer (5′-TGTAAATACTGCTTGCAGCC-3′) was annealed to poly(A)+ RNAs and cDNA was synthesized. After degrading RNA with RNase H, purified cDNA was tailed with dCTP and TdT. The dC-tailed cDNA was amplified with the anchor primer and a nested gene-specific primer (5′-GTGACCTGCAGAGATTAACC-3′). The double-stranded cDNA was subcloned into the pBluescript II vector and sequenced.

Bioinformatic analysis

The subcellular localizations of proteins were predicted using the PSORT program (Nakai & Horton, 1999). The secondary structures of proteins were predicted using the PSIPRED (McGuffin, Bryson & Jones, 2000), SAM-T99-2d (Karplus et al., 1999), and Profsec (Rost & Eyrich, 2001) programs. These programs were downloaded from the BioInfo MetaServer (http://bioinfo.pl/). The PSI-BLAST program (Altschul et al., 1997) was used for homology searches. Three iterations were used before full saturation was reached. The domain architecture of proteins was analyzed using the SMART tool (Letunic et al., 2002).

Immunoblot analysis

Proteins were extracted from testes using RIPA buffer and 10 µg of the sample was subjected to 8% or 10% (w/v) SDS-polyacrylamide gel electrophoresis (PAGE). The ORF of the testicular Runx2, which encodes a polypeptide of 106 aa, was fused in frame to glutathione-S-transferase (GST) using the pGEX vector. The plasmid was transfected into E. coli and transformed bacteria were lysed in sample buffer after induction with isopropyl-β-D-thiogalactoside. Two micrograms of protein was separated by SDS-PAGE. Proteins were transferred from the gel to a membrane and the blotted membranes were blocked with TBS-T buffer, which contained 20 mM Tris–HCl pH 7.4, 150 mM NaCl, and 0.1% (v/v) Tween 20. The primary antibody was anti-Runx2 serum, which was raised in rabbit using the amino-terminal 15 aa residues (MLHSPHKQPQNHKCG) of the murine testicular Runx2 as an antigen. In some cases, the antiserum was preabsorbed with an excess amount of antigen peptide. The secondary antibody was alkaline phosphatase-conjugated goat anti-rabbit IgG (Promega, Madison, WI, USA). The antibodies were diluted appropriately in TBS-T. Immunologically reacted products were detected using the BCIP/NBT Color Development Substrate Kit (Promega, Madison, WI, USA).

Immunohistochemistry of testicular preparations

To prepare frozen sections, testes from adult male C57BL/6J mice were cut into three pieces and fixed in Zamboni solution for 6 h at 4 °C with agitation. The tissues were immersed sequentially in 10% (w/v), 15%, and 20% sucrose solutions prepared in phosphate-buffered saline (PBS) for 2 h each at 4 °C; embedded in OCT compound (Miles Laboratories, Berkeley, CA, USA); and kept frozen at −80°C until use. The tissues were cryostat-sectioned (7-µm thick sections), air-dried, and post-fixed for 20 min in 4% (w/v) paraformaldehyde prepared in PBS.

The sections were then treated with methanol containing 0.3% (v/v) hydrogen peroxide for 15 min followed by PBS containing 3% (w/v) skimmed milk and 10% (v/v) goat serum for 30 min. In immunohistochemistry, the anti-serum was affinity-purified using a peptide (MLHSPHKQPQNHKCG)-linked Sepharose 4B column. The sections were incubated with appropriately diluted, affinity-purified anti-Runx2 antibody at 4 °C overnight, followed by biotinylated goat anti-rabbit IgG for 30 min. The primary antibody was detected using an ABC Kit (Vector Laboratories, Burlingame, CA, USA). The sections were post-stained with methyl green, and coverslips were mounted on glass slides.

Results

The testicular Runx2 is transcribed specifically in germ cells

To gain insight into the significance of the Runx2 transcript in testes, we first used W∕Wv mice and wild-type C57BL/6 mice. W∕Wv mice lack germ cells except spermatogonia because of mutations in the c-Kit gene (Kubota et al., 2009). RNA was extracted from testes and processed for Northern blot analysis (Fig. 1, Fig. S1). The Runx2 transcript was detected as a broad band of ∼1.8 kb length in wild-type testis, but not in W∕Wv testis (Fig. 1A, lanes 1 and 2). By contrast, Runx1 transcript (Fig. S1), PEBP2β∕CBFβ transcript (Fig. 1B, lanes 1 and 2) and β-actin transcript (Fig. S1) used as loading control, were detected in both RNA samples tested. This indicates that the testes-specific Runx2 transcript is expressed in germ cells, not in somatic cells in testes.

Figure 1 Northern blot analysis of Runx2 expression in testis.

RNA was prepared from 10-week-old W∕Wv and C57BL/6J mouse testes (lanes 1 and 2, respectively), from pachytene spermatocytes and round spermatids (lanes 3 and 4, respectively), and from 1-, 2- 3- and 4-week-old C57BL/6J mouse testes (lanes 5, 6, 7 and 8, respectively). The probes were cDNAs of the murine Runx2 (A) (a HindIII-NotI fragment corresponding to nt 282 through to nt 473 in NM_001146038.2) and PEBP2β∕CBFβ (B). The numbers alongside the gels show the sizes of the transcripts in kb.

Next, spermatocytes at the pachytene stage and spermatids were purified from a cell suspension prepared from wild-type testis, and then RNA analyzed. The Runx2 transcript was detected in both the spermatocyte and spermatid fractions (Fig. 1A, lanes 3 and 4). The differentiation of germ cells proceeds in a synchronous fashion immediately after birth; therefore, RNA was prepared from the testes of newborn mice and analyzed (Fig. 1A, lanes 5–8). A band of 1.8 kb was detected in 4 week-old testis, which largely contains germ cells at the spermatid stage. Thus, the testicular Runx2 was transcribed in germ cells at the spermatocyte and spermatid stages.

Determination of the structure of the testicular Runx2 transcript

Although the study by Ogawa et al., (2000) predicted the ORF of Runx2 to encode a 106-aa protein, the cDNA and aa sequences have not been published or registered in a public database. Only the junction sequence between exons 1 and 3 is available in the literature. Therefore, we decided to independently clone Runx2 cDNAs from a library prepared from murine testis. The nucleotide sequence that was determined from the obtained cDNA clones and 5′RACE is presented in Fig. 2A and has been deposited at NCBI (accession number, DQ458792). The AUG codons of Met1 and Met69 as well as the termination codon are indicated in bold.

Figure 2 The nucleotide sequence and the structure of testicular Runx2.

(A) The nucleotide sequence of the testicular Runx2 transcript (1,480 nucleotides) is shown. The two ATG codons represent the Met1 and Met69. The TGA represents the termination codon. The two forward slashes indicate the boundaries between exons 1U and 1D and between exons 1D and 3. The underlined sequence is a poly(A)-addition signal. (B) Comparison of open reading frames that are assigned to the testis- and bone-derived Runx2 transcripts is shown. The numbers represent the exon numbers and the boxes represent the coding regions.

We next compared this cDNA sequence with the murine genomic sequence of Runx2 (AB013129) (Xiao et al., 1998). The so-called exon 1 could actually be split into two small exons that are separated by an intronic sequence of 197 nucleotides. We tentatively designated these smaller exons as 1U (U for upstream) and 1D (D for downstream). The predicted coding region of each exon is shown in Fig. 2B. Exon 1U harbors the Met1 codon, whereas exon 1D includes the Met69 codon. Exons 1U and 1D are also transcribed in bone Runx2 (Xiao et al., 1998) and the ORF of bone Runx2 has been predicted ( NM_009820) (Ducy et al., 1997). The amino-terminal 87 aa residues are common to both the testicular and bone ORFs, whereas the carboxy-terminal 19 aa residues derived from exon 3 are unique to the testicular ORF.

Bioinformatic analysis suggests that the testicular Runx2 variant is authentic

The predicted aa sequence of the testicular Runx2 variant is shown in Fig. 3A. We performed bioinformatic analysis as a first step to examine the theoretical likelihood that this is an authentic protein. According to the PSORTII program, the Runx2 variant had a 43% probability of being a nuclear protein. Moreover, the variant appears to be a basic protein; out of 106 aa residues, 22 are basic and only six are acidic. Secondary structure assignment programs suggested that Runx2 has both α-helical and β-sheet structures (indicated by H and E, respectively; Fig. 3A). In particular, a stretch of 55 aa in the amino-terminus are folded into two distinct α-helices. A basic, nuclear protein with a high α-helical content is reminiscent of a histone. On the other hand, three carboxy-terminal Cys residues may adopt a globular structure with a sulfur bridge (indicated by green, Fig. 3A).

Figure 3 Bioinformatic analysis of the testicular Runx2 polypeptide.

(A) Secondary structures such as α-helices and β-sheets are indicated by H and E, respectively. The two Met and five Cys residues are indicated. The two forward slashes indicate the boundaries of the exons. (B) A homologous motif found in testicular Runx2 and Q7TPL8 proteins is shown. Identical or similar aa are indicated. (C) The domain architecture of the Q7TPL8 protein is shown. The circle and boxes represent the KRAB and C2H2-type zinc finger domains, respectively. The region showing similarity to testicular Runx2 is indicated by a red line.

The amino-terminal part of the Runx2 variant (from Met1 to Ser46) showed weak but significant similarity to a 48-aa sequence of a murine protein of unknown function that is expressed in the eye, namely, Q7TPL8/33942100 (SWISS and NCBI IDs, respectively; Fig. 3B). Q7TPL8 possesses features that suggest it is a transcription factor (Fig. 3C). At the amino-terminus, it contains a KRAB (Kruppel-associated box) domain that may function as a transcription-repression domain, and at the carboxy-terminus, it harbors eight zinc fingers that may function as nucleic acid-binding structures. The region of Q7TPL8 that shows similarity to Runx2 is located immediately before the stretch of zinc fingers. This feature suggests that the amino-terminal motif of 46 aa residues in Runx2 represents a functional domain. In addition, the testicular Runx2 also retains a stretch of 19 aa residues (from Met69 to Trp87) that corresponds to an important transcription activation domain in bone Runx2 (Thirunavukkarasu et al., 1998).

Taken together, these observations increase the probability that the testicular Runx2 is an authentic protein.

Met1 is used as an initiation codon in the testicular Runx2 variant

We next examined whether Met1 is indeed used as an initiation codon in testis. We generated an antiserum using the amino-terminal 15 aa of Runx2 (Met1 to Gly15) as an antigen. A protein extract was prepared from mouse testis and processed for immunoblot analysis. A clear band of approximately 11 kDa was detected (Fig. 4A, lane 1, indicated by an arrow), whereas inclusion of an excess amount of antigen peptide in the immunoreaction abolished this band (Fig. 4A, lane 2). Lysates from bacteria expressing a GST-Runx2 ORF fusion protein were run as controls in lanes 3 and 4, and GST-Runx2 was detected with the same antiserum. It is therefore highly likely that this ORF is expressed in testis as a polypeptide of 106 aa residues.

Figure 4 Immunoblot detection of Runx2 protein in testis and its localization in nuclei of spermatogenic cells dependent on seminiferous-stage.

(A) Immunoblot detection of Runx2 is shown. Lanes 1 and 2 contain protein from testes, whereas lanes 3 and 4 contain protein from GST-Runx2-transfected bacteria. The membrane was probed with anti-Runx2 serum, which was preabsorbed with the antigen peptide (lanes 2 and 4) or was not (lanes 1 and 3). The arrow and arrowhead indicate testicular Runx2 and the GST-Runx2 fusion protein, respectively. (B) Immunohistochemical staining of Runx2 protein in testis is shown. Testes from adult C57BL/6J mice were stained with the affinity-purified anti-Runx2 antibody and counterstained with methyl green. In b, the antibody was preabsorbed with the antigen peptide. The yellow scale bars correspond to 251 µm. Arrow in panel c: this staining was not disappeared when the antibody was preabsorbed with the antigen peptide. Arrow in panel d, h and i: nuclei of spermatogonia. (C) Enlarged view of positive staining in nuclei is shown. a: pachytene spermatocytes, b: round spermatids.

The testicular Runx2 is a differentiation stage-dependent nuclear protein

To verify that the truncated variant of Runx2 is expressed, we performed immunohistochemistry on frozen sections of testes (Fig. 4B). Panel a shows the staining pattern in mouse testis that was probed with the affinity-purified anti-Runx2 antibody raised against the Met1-to-Gly15 peptide. Positively stained cells were detected within seminiferous tubules. This immunostaining was specific for Runx2 because it completely disappeared when the antibody was preabsorbed with an excess amount of antigen peptide (panel b).

Interestingly, the distribution pattern of positive cells appeared to differ from tubule to tubule. Therefore, each seminiferous tubule containing positive cells were classified according to the differentiation stage. This was judged by the morphology of cells and nuclei (see panel c–i). Runx2 staining was detected in the following differentiation stages: late pachytene spermatocytes at stages VIII and X (panels c and d), diplotene spermatocytes at stage XI (panel e), cells of second meiotic phase at stage XII (panel f), and round spermatids of spermiogenesis phase 1, 2/3, and 5 at stage I, II/III, and V, respectively (panels g, h, and i). In short, the Runx2 variant was detected in various stages of differentiation, from late pachytene spermatocytes to round spermatids. It must be noted that detection of Runx2 protein by immunohistochemistry (Fig. 4B) and detection of the Runx2 transcript by Northern blot (Fig. 1) coincides in terms of germ cell specific expression and expression in pachytene spermatocytes to spermatids.

Finally, as seen in the enlarged view in Fig. 4C, positive staining was always restricted to the nuclei, indicating that the testicular Runx2 is a nuclear protein. The protein was detected as multiple foci in the nuclei of germ cells. The original picture of immunohistochemistry observed under differential interference contrast microscope can be seen as Figs. S2 and S3.

Discussion

This study showed that the testicular Runx2 is located in the nucleus, appears to be a basic protein, and has a predominantly α-helical conformation. These characteristics are somewhat reminiscent of histone proteins. The protein was detected in spermatocytes at the late pachytene and diplotene stages as well as in round spermatids. At the pachytene and diplotene stages, genetic information is exchanged between a pair of homologous chromosomes through homologous recombination. Thereafter, in post-meiotic and round spermatids, chromatins containing a haploid genome start preparing to remodel their structures. The testicular Runx2 protein detected in this study might be involved in the aforementioned processes. In this context, it is worth noting the histone variants that are expressed specifically in male germ cells. For example, a testicular variant of linker histone 1 is detected in pachytene spermatocytes and persists until the round spermatid stage (Brock, Trostle & Meistrich, 1980; Drabent et al., 1996). Likewise, TH2A and TH2B (testicular variants of the core histones H2A and H2B) and H3t (a testicular variant of the core histone H3) are expressed in round spermatids. It would be interesting to determine whether the testicular Runx2 is incorporated into nucleosomes and play roles in loosening their structures, as histone variants are suggested to do (Rathke et al., 2014). In any case, a single Runx2 gene can encode two different types of nuclear proteins, a previously defined transcription factor in bone and cartilage and a short testicular variant that lacks a Runt domain. The amino-terminal 87 aa residues shared by the testicular and bone Runx2 might exert distinct functions. Namely, the former might represent a histone-like protein whereas the latter likely represents a transcription activation domain.

Jeong et al. (2008) reported the expression of the Runx2 transcription factor in mice sperm. Their immunoblot analyses using a monoclonal antibody against Runx2 detected proteins of 47 and 65 kD in lysates from testes and sperm. Therefore, the observations of Jeong et al. (2008) appear to be substantially different from our results. However, it must be noted that our Northern blot analyses could detect, albeit faintly, bands larger than the major 1.8-kb band (see Fig. 1). If such larger transcripts harbor the Runt domain sequence, they probably encode the Runx transcription factor that is found in bone and T cells. Thus, the findings of Jeong et al. (2008) and the current study may not be contradictory.

Although this study used murine testes, the Runx2 gene is conserved among mammals, thus suggesting a possible extrapolation of our findings to other species. In this sense, we note that the two homologous transcripts are found in the NCBI database. They are AB573882.1 and AB573881.1, and are reported to be expressed in a periosteum tissue of rat. In these rat transcripts, splicing appears to skip exon 2 and fuse exons 1 and 3 as in the same way as a murine testicular Runx2 transcript. Whether a protein is expressed from this rat Runx2 transcript is not known at present.

Conclusion

In conclusion, a variant of Runx2 that differs from the bone isoform in its splicing is expressed in pachytene spermatocytes and round spermatids in murine testes, and encodes a histone-like, nuclear protein of 106 amino acid residues. Considering its nuclear localization and differentiation stage-dependent expression, Runx2 may function as a chromatin-remodeling factor during spermatogenesis.

Supplemental Information

Figure S1 Northern blot analysis of Runx family in testis

The Runx2 transcript was detected as a broad band of ∼1.8 kb length in wild-type testis, but not in W∕Wv testis (lanes 1 and 2). By contrast, Runx1 transcript (lanes 5 and 6) as well as PEBP2β/CBFβ transcript (lanes 9 and 10) and β-actin transcript used as loading control (lanes 13 and 14) were detected in both RNA samples tested. lane 1∼4: Runx2 (PEBP2αA), lane 5∼8: Runx1 (PEBP2αB), lane 9∼12: PEBP2 β/CBFβ, lane 13∼16: β-actin.

Click here for additional data file.

Figure S2 The original picture of immunohistochemistry observed under differential interference contrast microscope

Immunohistochemical staining of Runx2 protein in testis is shown. Testes from adult C57BL/6J mice were stained with the anti-Runx2 antibody and counterstained with methyl green.

Click here for additional data file.

Figure S3 The original picture of immunohistochemistry observed under differential interference contrast microscope

Immunohistochemical staining of Runx2 protein in testis is shown. Testes from adult C57BL/6J mice were stained with the anti-Runx2 antibody and counterstained with methyl green.

Click here for additional data file.

We are grateful to Dr. Y Nishina for providing a cDNA library from murine testis and to Dr. H Kawashima for valuable advice on antibody affinity-purification.

Additional Information and Declarations

Competing Interests

Author Contributions

Animal Ethics

Data Availability

Marcin Grynberg is an Academic Editor for PeerJ.

Satoru Kanto performed the experiments, analyzed the data, contributed reagents/materials/analysis tools, wrote the paper, prepared figures and/or tables, reviewed drafts of the paper, choice of the journal.

Marcin Grynberg, Yoshiyuki Kaneko and Jun Fujita performed the experiments, prepared figures and/or tables.

Masanobu Satake conceived and designed the experiments, wrote the paper, reviewed drafts of the paper.

The following information was supplied relating to ethical approvals (i.e., approving body and any reference numbers):

The Animal Facility of the Institute of Development, Aging, and Cancer, Tohoku University, an environmentally controlled and specific pathogen–free facility. Animal protocols were reviewed and approved by the Animal Studies Committee of the Tohoku University (relevant approval number: 2013-IDAC-Animal-013).

The following information was supplied regarding data availability:

NCBI: DQ458792.

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
