# Peer review of "A variant of Runx2 that differs from the bone isoform in its splicing is expressed in spermatogenic cells"

_PeerJ, doi:10.7717/peerj.1862_

## Round 0.1 · original submission · Minor Revisions

As shown in the comments, the authors showed a novel isoform of Runx2 (testicular Runx2). Although this finding is interesting, I agree with the the reviewers that there are minor concerns, which should be addressed before publication.

Reviewer 1 ·

Basic reporting

Line 189: “This indicates that Runx2 is expressed specifically in germ cells, not in somatic cells.” is confusing. Needs to be modified to indicate the expression of testis specific Runx2 transcripts.

Experimental design

Line 46: the intensity of the upper bands also was reduced with the addition of the antigen peptide. How do you know the upper bands are non-specific from this blot?

Figure 1: PEBP2beta was used for loading control. It is the Runx binding partner. Is the expression PEBP2beta constant during spermatogenesis?

The inclusion of the staining of TH2A, TH2B, H3t and/or a testicular variant of H1 is helpful to support the authors’ conclusions.

Validity of the findings

No Comments

Additional comments

Runx2 has been reported to express in several tissues including testis. In contrast to the Runx transcript found in other organs, the testicular Runx2 transcript is shorter because of premature termination and alternative splicing (Ogawa et al, 2000). Although Runx2 expression in testes was confirmed in vivo using Runx2 promoter-LacZ transgenic mice (Jeong JH et al., J Cell Physiol. 2008), actual sequences of the testicular Runx2 transcript and amino acid were still unknown. The manuscript by Kanto et al represents further analysis for the testis-specific Runx2. In particular, the authors cloned and sequenced the testicular Runx2 transcript. Based on the sequence data, they have generated the Ab against the testis-specific Runx2 and analyzed its expression patterns during testis development. Runx2 was expressed from late pachyten to round spermatid in the nucleus. In addition, the data base analysis revealed histone-like structure of the Runx2 short form. Thus, they proposed a model that Runx2 may function as a chromatin-remodeling factor during spermatogenesis.

Although actual biological importance and functions of testis-specific Runx2 are entirely unclear because Runx2-null sperms show no obvious phenotypes in sex determination, their motility and fertility, the authors have provided some answers for the long-standing questions. My only concern is the specificity of the Ab. Minor comments are also posted in the other sections.

Reviewer 2 ·

Basic reporting

Authors confirmed the testicular Runx2 transcript is detected in testes and translated starting from Met 1st and can be detected by newly developed antibody. Furthermore, immunohistochemical analyses revealed that the Runx2 polypeptide was located in the nuclei, and was detected in spermatocytes at the stages of late pachytene, diplotene and second meiotic cells as well as in round spermatids.

Over all, I agree the authors’ claims.

This reviewer has some minor concerns.
Authors used “the testicular Runx2” and “testicular Runx2” in a mixed manner.
Are there any strict rules for usage?

In line 182-183, as well as should be and?

Experimental design

Over all, I agree the authors’ experiments.

This reviewer has some minor concerns.
In Figure 1, CBF beta is not suitable for loading control.
Photos of EtBr stained gel, like ribosomal RNA in total RNA case, or suitable probed Northern results should be presented.

Validity of the findings

Over all, I agree validity of the findings.

Reviewer 3 ·

Basic reporting

No comments.

Experimental design

No comments.

Validity of the findings

The description "Testicular Runx2 is a histone-like protein" throughout this manuscript is based on the prediction of secondary structure (ll. 223-4). This prediction comes from the presence of several alpha-helical structures and basic amino acids in a nuclear protein. I wonder whether these features (several alpha-helix, high content of basic amino acids) is found only on testicular Runx2 isoform, not on bone Runx2. The authors should show the comparison of secondary structure between testicular Runx2 and bone Runx2, and also discuss about the possibility that bone Runx2 is (or not) a histone-like protein.

Additional comments

In this manuscript, the authors showed a novel isoform of Runx2 (testicular Runx2), which lacks Runt domain, shows differentiation stage-dependent expression. Although this finding is interesting, I have a concern, which should be addressed before publication.

---

## Round 0.2 · accepted · Accept

The authors did a great job with the review. The paper is very interesting and may offer new aspects in the evaluation of Runx2.